# A statistical approach to topological entanglement:
# Boltzmann machine representation of high-order irreducible correlations

Shi Feng,[1, *] Deqian Kong,[2] and Nandini Trivedi[1]

[1]*Department of Physics, The Ohio State University, Columbus, Ohio 43210, USA*
[2]*Department of Statistics, University of California, Los Angeles, California 90095, USA*
(Dated: November 9, 2023)

Strongly interacting systems can be described in terms of correlation functions at various orders. A quantum analog of high-order correlations is the topological entanglement in topologically ordered states of matter at zero temperature, usually quantified by topological entanglement entropy (TEE). In this work, we propose a statistical interpretation that unifies the two under the same information-theoretic framework. We demonstrate that the existence of a non-zero TEE can be understood in the statistical view as the emergent $n$th order mutual information $I_n$ (for arbitrary integer $n \geq 3$) reflected in projectively measured samples, which also makes explicit the equivalence between the two existing methods for its extraction – the Kitaev-Preskill (KP) and the Levin-Wen (LW) construction. To exploit the statistical nature of $I_n$, we construct a restricted Boltzmann machine (RBM) which captures the high-order correlations and correspondingly the topological entanglement that are encoded in the distribution of projected samples by representing the entanglement Hamiltonian of a local region under the proper basis. Furthermore, we derive a closed form which presents a method to interrogate the trained RBM, making explicit the analytical form of arbitrary order of correlations relevant for $I_n$. We remark that the interrogation method for extracting high-order correlation can also be applied to the construction of auxiliary fields that disentangle many-body interactions relevant for diverse interacting models.

## CONTENTS

## I. INTRODUCTION

High-order correlation is of great theoretical importance in many fields of physics. Its presence indicates an irreducible many-body correlation/interaction that cannot be explained using pairwise relations. Such correlations usually emerge in many-body interacting systems such as frustrated magnetic systems and complex networks [1–5]. A quantum analog of high-order correlations is the topological entanglement in topologically ordered states of matter at zero temperature [6–8]. The topological entanglement is usually quantified by an $O(1)$ constant $\gamma$ i.e. the topological entanglement entropy (TEE) as the universal contribution to the von-Neumann entropy $S(L) = \alpha L - \gamma$ of a subsystem with boundary length $L$ [9, 10]. Understanding the mechanism of such emergent phenomena in many-body systems involves methods and theories beyond the standard mean-field paradigm, which is the main workhorse for many interacting models. In recent years, there has been considerable activity on long-range entangled frustrated systems [11–13] and quantum many-body systems with topological order (TO) [14–17]. These systems feature strong interaction between local degrees of freedom, whereby mean-field or single-mode approximations fail to capture the essential physics that requires theories beyond Landau's symmetry-breaking paradigm, such as Kitaev spin liquid [15] and the toric code (TC) model [14].

In the study of many-body systems, a pivotal strategy

---

* E-mail:feng.934@osu.edu

involves analyzing their statistical properties through the lens of information theory, with a particular focus on quantum entanglement and entanglement entropy as its quantifier. This approach is especially pertinent in the context of frustrated quantum systems [18], where long-range topological entanglement is a defining characteristic. Unlike systems described by local order parameters, these systems exhibit patterns of entanglement woven into the many-body wave functions, with local degrees of freedom becoming collectively correlated through the high-order and long-range entanglement of emergent gauge fields intrinsic to topological order (TO). The long-range nature of this entanglement in TO is quantifiable via TEE. However, the extraction of TEE from long-range entangled lattice gauge theories is hindered by the limitations of local operation and classical communication protocols like quantum distillation, as referenced in [19–21]. Furthermore, TEE embodies a fundamentally non-dyadic many-body correlation, a fact underscored by the absence of correlation functions in the TC model involving fewer than four qubits [22, 23]. The challenge of detecting TEE and thus affirming the presence of TO necessitates a representation adept at capturing the non-local, high-order correlations between qubits. The existence of such high-order statistical correlations or interactions also aligns with the complexities faced in the analysis of complex networks and the depiction of many-body synergistic information [3, 24]. We point out that similar challenges are not unique to quantum systems but is also encountered in nuclear physics, where the interactions among multiple nucleons are a significant consideration [25, 26].

In this work, we exploit a statistical view point to high-order correlations and topological entanglement as its quantum analogue. We show such long-range high-order correlation between multi-partite patches in a topologically non-trivial subsystem can be described by high-order (quantum) mutual information $I_n$ with $n \geq 3$, with the Kitaev-Preskill (KP) and Levin-Wen (LW) constructions of TEE equivalent to $I_3$. There are two equivalent ways to interpret the TEE as $I_{n \geq 3}$. It can be viewed (1) as a property of the reduced density matrix $\rho$ of a subsystem, which may be assigned a fictitious quantum entanglement Hamiltonian according to $\rho \equiv \exp(-H)$ [27]; or (2) as a joint data distribution with high-order covariance from projective sampling on a topological quantum state [28, 29]. In analogy to the entanglement Hamiltonian in the previous case, such data distribution can be assigned to a fictitious *classical* Hamiltonian $\mathcal{H}$ consisting of Ising interactions of different orders. Then the presence of TEE can be interpreted as follows: there exists a set or sets of basis, whereby the projectively sampled data set $\{i\}$ on a subsystem with closed topology is a sample of the probability distribution $p_{\{i\}} \propto e^{-\mathcal{H}\{i\}}$ generated by $\mathcal{H}\{i\}$ with *high-order Ising interactions*. This is summarized in Table I. We will focus on the case (2) in our work, and will explain more details in Sec. IV regarding this argument. By the projective sampling on $n$ qubits within a skeletal subsystem with non-trivial topology in certain basis, we

TABLE I. Comparison between two interpretations of the topological entanglement: (1) topological entanglement as a property of the reduced density matrix $\rho$ of a subsystem; and (2) as a joint data distribution with high-order covariance of a Gibbs ensemble.

| | (1) | (2) |
|---|---|---|
| Measure | Density matrix $\rho$ | Gibbs distribution $p_{\{i\}}$ |
| Generator | Ent. Ham. $H$ $\rho = \exp(-H)$ | Ising Ham. $\mathcal{H}$ $p_{\{i\}} \propto \exp(-\mathcal{H})$ |
| Quantity | TEE | $I_{n \geq 3}$ of samples |
| Method | KP or LW construction | Sampling + RBM* |

show that the resultant projective samples exhibit effective Gibbs joint distributions that feature non-trivial $I_n$, establishing the possibility to witness TEE by exploiting classical statistical methods.

We further propose an energy-based statistical representation of such high-order correlation and TEE using restricted Boltzmann machine (RBM) widely used in machine learning. Notably, recent investigation on machine learning applications in quantum many-body physics has burgeoned [30–34]. It has been shown that by exploiting the generating power of these artificial neuron networks, the phase factors of a quantum state of various models can be faithfully captured, thus providing a network-based variational quantum many-body solver [30, 33]; and the representation power of RBM has lead to numerous application in the physics community [26, 35–42]. We show that the high-order information existing in the joint distribution of samples can be represented by a bipartite Ising model i.e. the two-body interacting network of an RBM, in which the effective many-body interactions between Ising spins in its visible layer provide a neural network representation of the high-order correlations. In order to interrogate the trained RBM and accurately extract the high-order information, we derive a closed analytical form of the effective $n$-body coupling of RBM relevant for $I_n$. This allows us to determine the existence of high-order correlation or TEE by sampling a subsystem, instead of reconstructing the complete wave function. We further remark that the RBM representation developed in this paper will also be useful for modeling many-body interactions using two-body interactions. Indeed, it was demonstrated in Ref. [26, 41] that RBM can be used to represent interacting models with three-body interactions. Hence the exact form of the effective $n$-body interaction of RBM provides a generic pathway to construct auxiliary fields which disentangle *arbitrary-order* interactions into two-body interactions.

This paper is organized as follows: Section II presents the statistical interpretation of TEE using high-order mutual information, its equivalence to existing constructions, and the formulation of TEE using arbitrary partitions of a subsystem. Section III presents projective measurements

on the exactly solvable TC model, which is then used to demonstrate the equivalence between TEE and the $I_n$ encoded in the joint probability distribution. Section IV discusses the structure of RBM (details of RBM are discussed in Appendix B), and presents the analytical representation of effective high-order interactions between spins $\sigma \in \{+1, -1\}$ in the visible layer, enabling the interrogation of the trained RBM (Representation for $\sigma \in \{0, 1\}$ is discussed in Appendix A). Section V shows a worked example of extracting high-order information by RBM in the joint probability distribution sampled from TC. Section VI concludes our results, and briefly discusses the potential application of our RBM construction in many-body interacting models.

## II. THE STATISTICAL INTERPRETATION OF TOPOLOGICAL ENTANGLEMENT

In this section, we present a statistical viewpoint towards TEE i.e. the statistical interpretation of TEE using high-order mutual information. We start from the existing KP construction and LW construction and show their equivalence to the third-order mutual information $I_3$. Then we generalize this information-theoretic argument to arbitrary order $I_n$. This statistical formulation then naturally establishes the possibility of the representation via energy-based statistical network to be discussed in the following sections. By exploiting the property of $I_n$, we show a generic construction of TEE using an arbitrary $n$-partite subsystem at the end of the section.

### A. Kitaev-Preskill Construction

TO is characterized by global long-range entanglement. For gapped systems the von-Neumann entropy of a subsystem density matrix $\rho_A$ for the ground state scales as

$$S(\rho_A) = \alpha|\partial A| - \gamma + O(1/|\partial A|) . \tag{1}$$

The TO is reflected in the topological entanglement entropy (TEE) $S_{\text{topo}} \equiv -\gamma$ [9, 10, 43]. In the Kitaev-Preskill (KP) construction, $\gamma$ is extracted by a tripartite disk $A \cup B \cup C$ within the 2D lattice according to

$$\begin{aligned} S_{\text{topo}} &= S(\rho_A) + S(\rho_B) + S(\rho_C) \\ &\quad - S(\rho_{AB}) - S(\rho_{BC}) - S(\rho_{AC}) + S(\rho_{ABC}) \end{aligned} \tag{2}$$

whereby the linear combination is engineered in a way such that the area-law contribution is canceled out. $\gamma$ is related to the quantum dimension of anyonic charges in the medium by a topological quantum field theory (TQFT) whereby $\gamma = \log \mathcal{D}$, $\mathcal{D} = \sqrt{\sum_a d_a^2}$. Besides the relation with TQFT, recently there have been pioneering works in relating the information-theoretic framework, i.e. the quantum (conditional) mutual information with TO [44–46] resulting in non-trivial bounds and proofs.

We propose to view TO from the information-theoretic point of view as an emergent statistical synergy [24] due to the underlying gauge structure, which provides more direct intuition and possible detection of TO by quantum sampling in relevant computational and experimental platforms such as tensor network and Rydberg atom arrays. The simple interpretation that is central to our argument is that the topological entropy can be written in the form of (quantum) conditional mutual information:

$$S_{\text{topo}} = I(A : B) - I(A : B|C) \tag{3}$$

where $I(A : B)$ and $I(A : B|C)$ are quantum mutual information and quantum conditional mutual information, respectively defined by

$$I(A : B) = S(\rho_A) + S(\rho_B) - S(\rho_{AB}) \tag{4}$$
$$I(A : B|C) = S(\rho_{AC}) + S(\rho_{BC}) - S(\rho_C) - S(\rho_{ABC}) \tag{5}$$

This is consistent with the KP construction defined in Eq. 2 if subsystems are properly chosen. To be specific, the mutual information $I(A : B)$ quantifies the amount of shared information between $A$ and $B$; whereas the conditional mutual information $I(A : B|C)$ quantifies the amount of shared information between $A$ and $B$ *given that C is known* (e.g. by a local projection on a quantum state), and is able to include the irreducible tripartite information which is also known as synergistic information in the field of statistics [24]. While a non-zero $I(A : B|C)$ is capable of detecting the existence of synergy that information shared between two subsystems could be influenced by a third, it should be noted that it could confuse a trivial bipartite mutual information and the intrinsic synergy if there exists some shared information between $A$ and $B$ independent of $C$. A trivial case is where $A$ and $B$ is completely disjoint from $C$ thus $I(A : B|C) = I(A : B)$. Therefore, to remove such trivial cases, one must compare $I(A : B|C)$ against $I(A : B)$, and look into $I_3(A : B : C)$ defined in Eq. 3 or Eq. 6. It is only in the simplest case where $I(A : B) = 0$ that a non-zero $I(A : B|C)$ alone suffices to determine the synergistic information (as we are to demonstrate in the coming section, this is exactly the case of the LW construction).

If the tripartition of a subsystem is topological non-trivial e.g. shown in Fig. 1(b,c) and the ground state is topologically ordered, the resulting $I_3$ will equal to $S_{\text{topo}}$ [47, 48]. In contrast, for a topologically trivial geometry e.g. shown in Fig. 1(a), $I_3 = 0$ is always true for gapped systems since $I(A : B) = 0$ and $(A : B|C) = 0$; and it is true regardless of the order of $A, B$ and $C$ because $I_3(A : B : C)$ is symmetric under permutation of variables.

Notably, this interpretation coincides with that of the third order mutual information $I_3^c$ for classical random variables and is related to the measure of frustration and synergy [1, 24]. In the probabilistic context, given three random variables $A, B, C$ generated from a classical ensemble or sampling of quantum density matrix, we can

define $I_3^c$ for classical random variables as

$$I_3^c(A : B : C) = I^c(A : B) - I^c(A : B|C) \qquad (6)$$

where the mutual information and the conditional mutual information are defined respectively as

$$I^c(A : B) = \sum_{a,b} p(a,b) \log \frac{p(a,b)}{p(a)p(b)} \qquad (7)$$

$$I^c(A : B|C) = \sum_{a,b,c} p(a,b,c) \log \frac{p(c)p(a,b,c)}{p(a,c)p(b,c)} \qquad (8)$$

hence the $I_3^c(A : B : C)$ in Eq. 6 can be expressed compactly as

$$I_3^c(A : B : C) = -\sum_{a,b,c} p(a,b,c) \log \frac{p(a,b,c)p(a)p(b)p(c)}{p(a,b)p(b,c)p(a,c)} \qquad (9)$$

Note this definition is symmetric under permutations of indices of $A, B, C$. A negative $I_3^c$ is intimately related to the so-called synergistic information or interaction information, that is, the intrinsic many-body and non-dyadic correlation that cannot be reduced to pair-wise relations [1, 24]. A negative value of $I_3^c$ is then interpreted as a statistical situation in which the knowledge of any one of the three variables, $A, B$ and $C$, enhances the correlation between the other two. Such statistical intuition remains valid in the quantum many-body cases, whereas the many-body quantum correlation is attributed to the non-local entanglement in a topologically ordered wave function. Indeed, in pure lattice gauge theories or integrable models where gauge sector is disjoint from matter (such as Kitaev spin liquid), by choosing a set of basis whereby the Wilson loops are explicitly constructed, the quantum samples taken under these basis can be interpreted using the classical $I_3^c(A : B : C)$, and relevant statistical methods like restricted Boltzmann model can be straightforwardly applied on subsystems thereof.

This formulation of topological entropy can be applied to both classical and quantum context. Indeed, even for classically frustrated spins, the thermal fluctuations exhibit topological entropy and the synergy with negative $I_3$ [1, 49]. In the context of topologically ordered quantum matter, the non-local constraint e.g. plaquettes operators and Wilson loops in Toric code and Kitaev spin liquids intuitively resemble the aforesaid synergy. Indeed, as we are to show in detail, the topological entanglement entropy is equivalent to a quantum case of third order mutual information that indicates a synergy of quantum fluctuations which is present at zero temperature. This rethinking of topological entropy allows us to unify classically frustrated systems and the TO of quantum frustrated systems in the same information-theoretic framework.

### B. Levin-Wen construction

Indeed the Levin-Wen (LW) construction is equivalent to the KP construction, thus to $I_3$; and the aforemen-

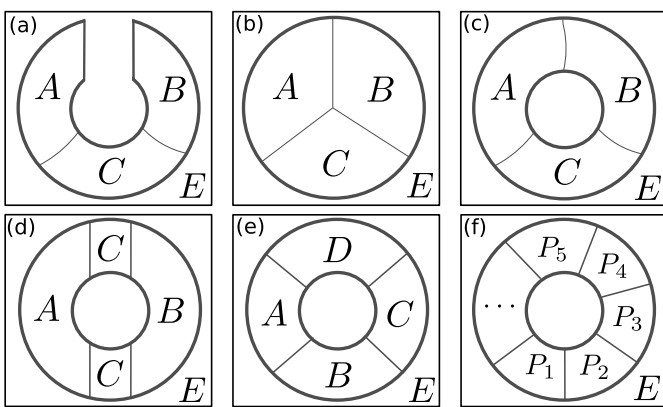

FIG. 1. Different partition schemes of annulus or disk. The partition in (a) is topologically trivial and has no TEE between the three partitions. (b) A tripartite disk used in the KP construction. (c) A tripartite annulus equivalent to Kitaev-Preskill construction. (d) A tripartite annulus used in the Levin-Wen construction. (e) A quadrupartite annulus. (f) An $n$-partite annulus. $E$ denotes environmental degrees of freedoms with respect to the annulus or disk.

tioned statistical interpretation of TEE remains valid as well. The LW construction extracts TEE using the partition shown in Fig. 1(d) and the following linear combination of entropies

$$S_{\text{topo}} = S(\rho_{ABC}) - S(\rho_{AC}) - S(\rho_{BC}) + S(\rho_C) \qquad (10)$$

It is equivalent to negative conditional mutual information $-I(A : B|C)$ which measures the shared information between $A, B$ conditioned on $C$, and a non-zero value indicates the existence of the long-range entanglement as a global constraint. It is necessarily non-positive due to the strong subadditivity inequality. Indeed, the LW construction defined above is also equivalent to $I_3$. This would be clear if we realize that $I_3$ for the geometry of Fig. 1(d) is $I_3(A : B : C) = I(A : B) - I(A : B|C) = I(A : B) + S_{\text{topo}}$. For a large enough $C$ whose length scale is larger than correlation length i.e. $\xi/|\partial C| \sim 0$, the Hilbert subspaces of $\rho_A$ and $\rho_B$ are disjoint, hence

$$S(\rho_{AB}) = S(\rho_A \otimes \rho_B) = S(\rho_A) + S(\rho_B)$$
$$\Rightarrow I(A : B) = 0 \qquad (11)$$

for the partition of Fig. 1(d). Combining Eq. 11 with Eq. 3 or Eq. 6 gives exactly the LW construction in Eq. 10. Nevertheless, if $\xi/|\partial C|$ is not negligibly small, it would be more accurate to include the term $I(A : B)$ which removes the residual non-topological information caused by finite range correlation. We would like to mention that there are also other construction of topological entropy that are statistically equivalent to the third order mutual information $I_3$ such as the multi-partite entanglement in the context of holographic theory [22, 50–53], which we do not elaborate in this paper.

## C. Higher order $I_n$ of TO is equivalent to $I_3$

In this section we show that, for a TO with emergent gauge theory, the $n$-th order quantum mutual information $I_n$ ($n > 3$) is equivalent to $I_3$. According to the aforementioned idea of higher order mutual information, we can always generate higher order constructions as descriptors of higher order irreducible many-body correlation. The generic $n$-th order quantum mutual information can be expressed recursively as

$$I_n(P_1 : \cdots : P_n) = I_{n-1}(P_1 : \cdots : P_{n-1}) - I_{n-1}(P_1 : \cdots : P_{n-1}|P_n) \quad (12)$$

where the conditional mutual information satisfies

$$I_{n-1}(P_1 : \cdots : P_{n-1}|P_n) = I_{n-1}(P_1 : \cdots : P_{n-1}P_n) - I_{n-1}(P_1 : \cdots : P_{n-2} : P_n) \quad (13)$$

Similar to $I_3$, Eq. 12 can be perceived as an $n$-body irreducible correlation, or $n$-body synergy that cannot be reduced to any correlations between fewer degrees of freedom. Indeed, it is intuitively clear that TEE is a direct consequence of the Gauss law of an emergent gauge theory, thus encodes such synergy between all degrees of freedom that live on a closed Wilson loop. Without loss of generality, let us assume a gapped TO whose correlation length is negligibly small compared to the sizes of subsystems. Note that for $n \geq 3$, the first term on the right-hand side of Eq. 12 and the second term on the right-hand side of Eq. 13 must be zero since the union of partitions in the parenthesis does not fill up an annular region i.e. is not subjected to gauge constraint; and the small correlation length guarantees there is no statistical correlation between subsystems that do not share a common boundary. Hence, we have

$$I_n(P_1 : \cdots : P_n) = -I_{n-1}(P_1 : \cdots : P_{n-1}P_n) \quad (14)$$

Following such recursion

$$I_n(P_1 : \cdots : P_n) = (-1)^{n-1} I_3(P_1 : P_2 : P_3 \cdots P_n) \\ \equiv (-1)^{n-1} S_{\text{topo}} \quad (15)$$

Hence TEE is equivalent to $I_n$ ($\forall n \geq 3$) up to a sign. As a concrete example, we demonstrate that the construction in Eq. 12 with $n = 4$, i.e. the 4th order mutual information with the partition shown in Fig. 1(e), is equivalent to $I_3$. For a quadrupartite annular disk $A \cup B \cup C \cup D$, $I_4(A : B : C : D)$ is defined as

$$I_4(A : B : C : D) = I_3(A : B : C) - I_3(A : B : C|D) \quad (16)$$

where $I_3(A : B : C)$ is necessary zero since $A \cup B \cup C$ does not form a closed topology and is thus equivalent to the conditional entropy of a quantum Markov state. The third order conditional mutual information $I_3(A : B : C|D)$ can be expanded in a non-conditional form as

$$I_3(A : B : C|D) = I_3(A : C : BD) - I_3(A : C : D) \quad (17)$$

where we have exploited the permutation symmetry of $I_n(P_1 : \cdots : P_n)$ to switch $C$ and $B$. Note in the given geometry of Fig. 1(e) $I_3(A : C : D)$ is necessarily zero [45], in the same way that $I_3(A : B : C) = 0$. Therefore, for a quadrupartite annular disk $A \cup C \cup B \cup D$, we have

$$I_4(A : B : C : D) = -I_3(A : C : BD) \quad (18)$$

the latter is the same as a tripartite annular disk $A \cup C \cup BD$, thus $I_4$ is reduced to the LW construction of $I_3$ as shown in Fig. 1(d). It is then trivial to extend to proof to $n$th order by induction. Indeed, this is in accordance with the irreducible correlation $C_\rho^{(k)}$ which measures the (quantum) information that is contained in $k$ parties yet absent in $(k-1)$ or less [54]; and, in particular, both KP and LW constructions can be understood as the third order irreducible correlation $C_\rho^{(3)}$ [55]. Generally, in systems where the correlation length is not negligibly small compared to the distances between subsystem partitions, we can write down the complete form of $I_4$ by Eq. 12 and Eq. 13 to extract topological entanglement entropy $S_{\text{topo}}^{(4)}$ using the quadripartite disk in Fig. 1(e):

$$-S_{\text{topo}}^{(4)} = S(\rho_A) + S(\rho_B) + S(\rho_C) + S(\rho_D) \\ - S(\rho_{AB}) - S(\rho_{BC}) - S(\rho_{AC}) - S(\rho_{BD}) \\ - S(\rho_{AD}) - S(\rho_{CD}) \\ + S(\rho_{ABD}) + S(\rho_{BCD}) + S(\rho_{ACD}) + S(\rho_{ABC}) \\ - S(\rho_{ABCD}) \quad (19)$$

which is the quadripartite analogue to the tripartite KP construction. It is readily to check that all boundary contributions cancel with each other. Higher order constructions can be represented by the same token. For a generic $n$-partite partition shown in Fig. 1(f), we have $\forall n \geq 3$:

$$S_{\text{topo}}^{(n)} = (-1)^{n-1} I_n = \sum_{i=1}^{n} (-1)^{i+n} \sum_{k_1, \cdots, k_i} S(\rho_{(P_{k_1} \cdots P_{k_i})}) \quad (20)$$

as a generic recipe for extracting TEE in an $n$-partite subsystem.

## III. STATISTICS OF ENTANGLEMENT SAMPLING

The intimate relation between high-order irreducible mutual information (or information synergy) and non-local correlation in topologically ordered systems naturally calls for a probabilistic interpretation in quantum models. In this section we present the sampling process and an example by TC which harbors a $Z_2$ gauge theory.

## A. Joint distribution from projective sampling

In particular, we focus on the entanglement of "skeletal" regions in lattice models, which in 2D lattices are lines with no volume. The advantage of such partition is that for gapped systems it requires the minimum number of qubit samples in order expose the topological structure of the wave function, such as a Wilson loop operator; and $I_n$ can thereby be interpreted as the $n$-th order mutual information between $n$ sampled qubits. By exploiting the inherent probabilistic nature of quantum wavefunction, we can define the probability distribution given a set of projective measurements $\mathcal{P}_i$ [29]

$$p_i = \text{Tr}\left(\mathcal{P}_i^\dagger \mathcal{P}_i \rho\right), \quad \rho' = \frac{\mathcal{P}_i \rho \mathcal{P}_i^\dagger}{\text{Tr}\left(\mathcal{P}_i^\dagger \mathcal{P}_i \rho\right)} \qquad (21)$$

where $i$ denotes a state in certain complete computational basis $\{\sigma_i^{\alpha_i} | i = 1 \cdots n, \alpha \in \{0, x, y, z\}\}$ and $\rho'$ is the resultant density matrix after the measurement. Assume each $\mathcal{P}_i$ is a local projector, we conduct sampling on skeletal partition of the lattice i.e. regions (lines) that have no volume [56] but with non-trivial topology as closed loops. Then in an n-site subsystem one can consider the probability in the auto-regression form

$$p(\sigma_1^{\alpha_1}, \cdots, \sigma_n^{\alpha_n}) = \prod_{i=1}^{n} p(\sigma_i^{\alpha_i} | \vec{\sigma}_{<i}) \qquad (22)$$

where $\vec{\sigma}_{<i} \equiv \{\sigma_j^{\alpha_j} | j < i\}$. It should be pointed out that in principle one can simply project the state into a definite many-body basis without doing a series of local projection, however, the local projections easily allow the study on local density matrices and also make it amenable to tenor network methods like density matrix renormalization group and matrix product states. Eq. 22 can be treated as a classical probability distribution for any normalized wavefunction. Note that even though it is formalized in a classical probability, quantum information are nevertheless accessible by shuffling the local computational basis; and since the probability $p(\sigma_i^{\alpha_i} | \vec{\sigma}_{<i})$ is determined by the projective measurements conditioned on the totality of previous measurements, the sampled data is in general able to reflect the entanglement structure of $\rho$ according to Eq. 21. Assume that $\{\sigma_i\}$ are gauge field degrees of freedoms as is required by the TO, where local contractable Wilson loops is constraint to have zero total flux, and any other operators that do not form closed loops, i.e. those that are not gauge-invariant, are not subjected to such constraint. Then the joint probability in Eq. 22 will exhibit a strong synergy if $(\sigma_1^{\alpha_1}, \cdots, \sigma_n^{\alpha_n})$ forms a Wilson loop whereby strong fluctuations are accompanied by many-body constraints; and weak synergy otherwise.

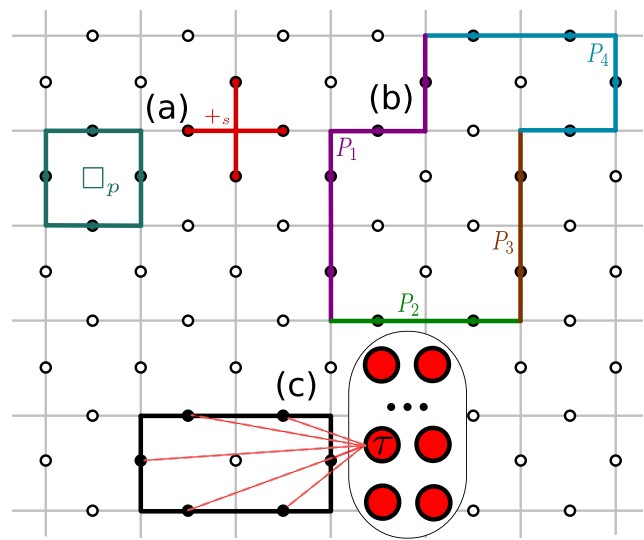

FIG. 2. The Toric code lattice, with subsystem partitions as skeletal regions. (a) The two mutually commuting terms of the TC Hamiltonian in Eq. 23. (b) A skeletal subsystem consisting of four partitions $P_1, \cdots, P_4$, which corresponds to the topology of Fig. 1(e) whereby TEE can be described by the mutual information $I_4$. (c) An RBM network representation for the sample distribution of a local skeletal region. The red nodes belong to the hidden layer of the RBM, the projectively measured spins in the TC lattice are treated as the visible nodes of the RBM. For simplicity we only draw the couplings in pink dashed lines between one hidden node and visible nodes within the region of interest. If measurements are projected on the $z$ axis, the RBM will be able to discern an effective fourth order interaction that is responsible for the $I_4$ encoded in the plaquette Wilson loop of TC.

## B. Pure $Z_2$ gauge theory

In this section, we demonstrate the equivalence between the following three key concepts: (i) zero-flux constraint for Wilson loops (ii) topological entanglement entropy and (iii) higher order mutual information $I_3$ in the TC model as a pure $Z_2$ gauge theory by classical probabilistic treatment. The simplest macroscopic model that realizes a pure $Z_2$ gauge theory and TO is the Toric code model, where the local energy density is given by mutually commuting stabalizers that are responsible for the non-local loop operators. Let us start with a pure $Z_2$ gauge theory, which can be realized by macroscopic Hamiltonian by e.g. Toric code model

$$H_{\text{TC}} = -J_{\text{TC}} \left[ \sum_s A_s + \sum_p B_p \right] \qquad (23)$$

with $A_s$ and $B_p$ given by

$$A_s = \prod_{i \in +_s} \sigma_i^x, \quad B_p = \prod_{i \in \square_p} \sigma_i^z \qquad (24)$$

as shown in Fig. 2(a). Its ground state can be exactly constructed by superposing all gauge-equivalent wave

functions: $|\Psi\rangle \propto \prod_s (1 + A_s) |\Psi_0\rangle$, where $|\Psi_0\rangle$ can take the form of any $z$-basis state that satisfies $\prod_{i \in \square_p} \sigma_i^z = 1$. Let us consider four-site plaquette subsystem consisting of link variable $\sigma_\square \equiv \{\sigma_1, \sigma_2, \sigma_3, \sigma_4\}$. The topological nature can be reflected in the fact that the product of four $\sigma^z$ ($\sigma^x$) that reside in a plaquette (star) must be $+1$ for the ground state wavefunction; and that the gauge operator enforces the superposition of all gauge configurations that satisfies the aforesaid constraint. The ground state wave function can hence be written as

$$|\Psi\rangle \propto \sum_{\{\sigma_\square\}} |\sigma_\square\rangle \otimes |\overline{\sigma_\square}\rangle \qquad (25)$$

up to a normalizing factor, where $|\overline{\sigma_\square}\rangle$ denotes the gauge configuration of links complementary to the $\sigma_\square$. The summation is over the configuration $\{\sigma_\square\}$ of subsystem, which determine the set of configurations in the environment. Due to the zero-flux constraint in the ground state, there are three degrees of freedom which fluctuate independently, hence, the normalized reduced density matrix takes the form

$$\rho_\square = \text{Tr}_{\overline{\sigma_\square}}(|\Psi\rangle \langle \Psi|) = \frac{1}{2^3} |\sigma_\square\rangle \langle \sigma_\square| \qquad (26)$$

which immediately gives the entropy $S = L \log 2 - \log 2$, with $L = 4$ and the second term $S_{\text{topo}} = -\log 2$. The simplicity of $\rho_\square$ of TC makes it an ideal and minimal platform to test the statistical approach. To apply a statistical measure of topological entanglement, we use the projective measurement which produces classical probabilities. Given a mixed state $\{p_i, |\psi_i\rangle\}$, the (reduced) density matrix is defined by $\rho = \sum_i p_i |\psi_i\rangle \langle \psi_i|$. A projetive meaurement by projector $\mathcal{P}_i$ gives the outcome $i$ with probability $p_i$, and $\rho$ collapses into $\rho'$ as discussed in Eq. 21.

Note that the gauge constraint in $Z_2$ TO states that any contractable loops must have zero flux in the ground state expectation, hence, under $z$ basis, in each set of samples the four spins must multiply to one. Equivalently, fixing one of the four spins by a projection into $|\uparrow\rangle$, projective measurements on the remaining three $\sigma_i$ should give a dependent sample distribution. This allows us to write it in terms of the tripartite form, which immediately gives a non-zero synergistic information as the topological entanglement. To see this, we calculate the (conditional) mutual information of $\{\sigma_1, \sigma_2, \sigma_3\}$ in the classical information context given a fixed $\sigma_4 = +1$:

$$I(\sigma_1 : \sigma_2 | \sigma_4 = +1) = \sum_{\sigma_1, \sigma_2} p(\sigma_1, \sigma_2) \log \frac{p(\sigma_1, \sigma_2)}{p(\sigma_1) p(\sigma_2)} \quad (27)$$

$$\begin{aligned} I(\sigma_1 : \sigma_2 | \sigma_3 ; \sigma_4 = +1) &= \sum_{\sigma_1 \sigma_2 \sigma_3} p(\sigma_1, \sigma_2, \sigma_3) \\ &\times \log \frac{p(\sigma_3) p(\sigma_1, \sigma_2, \sigma_3)}{p(\sigma_1, \sigma_3) p(\sigma_2, \sigma_3)} \end{aligned} \quad (28)$$

It is straightforward to evaluate these equations noting that there are only a few choices of $\{\sigma_1, \sigma_2, \sigma_3\}$ given the zero-flux constraint. Here we can directly write down the marginal probabilities. Due to the gauge symmetry, all valid gauge configurations share the same probability weight, hence we have

$$p(\sigma_i = \pm 1) = \frac{1}{2}, \; p(\sigma_1 = \pm 1, \sigma_2 = \pm 1) = \frac{1}{4} \qquad (29)$$

$$p(\sigma_1 = \pm 1, \sigma_2 = \pm 1, \sigma_3 = \pm 1) = \frac{1}{4} \qquad (30)$$

These immediately gives

$$I(\sigma_1 : \sigma_2 | \sigma_4 = +1) = 0, \qquad (31)$$
$$I(\sigma_1 : \sigma_2 | \sigma_3 ; \sigma_4 = +1) = \log 2 \qquad (32)$$

thus the third order mutual information which coincides with TEE:

$$I_3(\sigma_1 : \sigma_2 : \sigma_3 | \sigma_4 = +1) = -\log 2 = S_{\text{topo}}(\text{TC}) \quad (33)$$

By the same token we would arrive at the same $\log 2$ with the fixed $\sigma_4 = -1$. This is also consistent with that obtained by LW construction on a skeletal region [56]. Hence in the chosen basis, the topological entanglement inside the plaquette Wilson loop can be treated as a classical statistical problem where a negative third order mutual information $I_3$ is indicative of the existence of TEE; and this holds true for larger loops with negative $I_n$. Note that the equal-weight superposition between gauge configuration, thus the presence of the gauge operator $\sum_s \prod_{i \in +_s} \sigma_i^x$ in the Hamiltonian, is essential in deriving the above result. Its absence will lead to a product state that trivially satisfies the constraint $\langle \prod_\square \sigma_i^z \rangle = 1$ at zero temperature without fluctuation in the samples of projective measurements.

This toy example also clearly showcases the gauge constraint as an essential piece that give rise the synergistic information, in the same way it is needed in the conventional derivation of TEE in previous references. In the statistical point of view, the TEE is equivalent to the intrinsic many-body correlation that cannot be reduced to several correlations of pairs of qubits. The key role played by the gauge constraint can also be reflected in another kind of subsystem partition: assume a subsystem whereby the constituent four qubits are colinear, thus the gauge constraint does not interfere. In this case $I_3 = 0$ since no synergy would emerge in absence of gauge constraint; this is also consistent with Ref. [45] where the authors showed the quantum conditional mutual information vanishes for subsystems with colinear topology. Hence, to witness $S_{\text{topo}}$ or $I_n$, we must partition a subsystem into a topology such that the gauge constraint is present, such as the skeletal loop shown in Fig. 2(b). Nevertheless, we would like to point out that it is still possible to extract TEE in certain fine-tuned scenarios using only two-point correlations if there are particles in additional quantum sectors that are coupled to the emergent gauge field, whereby the information of gauge sector is imprinted into the two-point correlators of the matter sector [23].

## IV.   STATISTICAL REPRESENTATION BY RESTRICTED BOLTZMANN MACHINE

The goal is to capture the irreducible many-body correlation or the synergistic information present in $I_n$ in the probability distribution of the samples generated from projective measurements on a potentially topologically ordered pure state. The correlation of a state $\Psi$ is encoded in the reduced density matrix of a subsystem $S$, which can be formally associated to a entanglement Hamiltonian $\mathcal{H}(\sigma \in S)$ [27, 57] (we use calligraphic $\mathcal{H}$ here in contrast to physical Hamiltonian):

$$\rho_S = \mathrm{Tr}_{\bar{S}} |\Psi\rangle \langle\Psi| \equiv e^{-\mathcal{H}(\sigma \in S)} \qquad (34)$$

where $\bar{S}$ denotes the complement set of degrees of freedoms of $S$. In this formulation, the many-body correlation or topological entanglement is encoded in an interacting Hamiltonian $\mathcal{H}(\sigma \in S)$, where the $n$-body correlation/entanglement of the density matrix can be reflected in the $n$-body interaction of the entanglement Hamiltonian. In a pure gauge theory like TC, it suffices to include only diagonal elements in the $\sigma^z$ basis, and Eq. 34 is reduced to a Boltzmann form $p = e^{-\mathcal{H}}/Z$.

Representing the correlation information by a interaction model $\mathcal{H}(\sigma \in S)$ in Eq. 34 requires a statistical network that is able to capture the high order correlation in the data distribution from samples using manageable amount of coupling parameters. As the universal approximator, deep neural networks are in general good at capturing non-local and high-order information such as TO that exhibit $I_n$ for large $n$. However, in order to retain analytical tractability, we choose to apply the RBM as a representation model for the projectively sampled data from a quantum state. RBM is defined in a bipartite Ising lattice whereby only one subset of spins $\sigma$ are physical (visible) whereas auxiliary spins $\tau$ in the other subset are deemed unphysical (hidden). It is essentially a generalized auxiliary field representation of coupled binary spins where the arbitrarily high order interactions between $\sigma$ can be represented by at most second order interaction between $\sigma$ and $\tau$. For detailed discussions of RBM we refer readers to the Appendix as well as Ref. [58].

### A.   Many-body interaction via two-body interaction

In this subsection we explain the intuition that a two-body interacting model like RBM can be used to construct many-body correlation/interaction. It has been proven that with a sufficiently large number of hidden nodes, any probability distribution can be well approximated by RBM [59]. The essence of RBM is the representation of an effective model with many-body interaction using a model with redundant degrees of freedoms which are coupled only by two-body interactions. To make this point explicit, Let $H_\tau$ be the Hamiltonian of the hidden nodes, which can simply take the form of Pauli matrices $\tau$ in presence

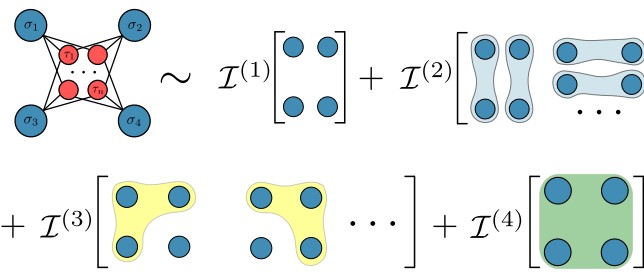

FIG. 3. An illustration of the effective interaction of visible nodes in an RBM, where $r$ blue circles that are included in the same shaded region enjoy an $r$-body interaction $\mathcal{I}^{(r)}$. The four blue circles denote the visible nodes $\sigma$, and the $n$ red ones denote the hidden nodes $\tau$. The two-body couplings between $\sigma$ and $\tau$ give effective higher order interactions between $\sigma$ when the hidden layer is traced out.

of magnetic fields; and let $H_\sigma$ be the Hamiltonian of the visible nodes that are coupled by less than or equal to two-body interactions, and $H_{\sigma\tau}$ the Hamiltonian of the two-body interactions between the visible and the hidden nodes. The generic Green function $G$ of the whole system is given by $(E - H)G = I$, in block matrix form it can be written as

$$\begin{pmatrix} E - \mathcal{H}_\tau & -\mathcal{H}_{\sigma\tau} \\ -\mathcal{H}_{\sigma\tau}^\dagger & E - \mathcal{H}_\sigma \end{pmatrix} \begin{pmatrix} G_\tau & G_{\sigma\tau} \\ G_{\tau\sigma} & G_\sigma \end{pmatrix} = \begin{pmatrix} 1 & 0 \\ 0 & 1 \end{pmatrix} \qquad (35)$$

This immediately gives

$$\left\{ E - \left[ \mathcal{H}_\sigma + \mathcal{H}_{\sigma\tau}(E - H_\tau)^{-1} H_{\sigma\tau} \right] \right\} G_\sigma = I \qquad (36)$$

which means the Green function of the visible nodes is

$$G_\sigma = \frac{1}{E - (\mathcal{H}_\sigma + \Sigma)} \qquad (37)$$

with the self energy $\Sigma$ given by

$$\Sigma = \mathcal{H}_{\sigma\tau}(E - \mathcal{H}_\tau)^{-1}\mathcal{H}_{\sigma\tau} \qquad (38)$$

It is $\Sigma$ which involves higher order interaction in the perturbation expansion and gives the representation power of RBM. Therefore we can identify an effective Hamiltonian of the visible nodes under the influence of the hidden nodes. The effective Hamiltonian is simply

$$\mathcal{H}_\sigma^{\mathrm{eff}}(E) = \mathcal{P}_\sigma(\mathcal{H}_\sigma + \Sigma)\mathcal{P}_\sigma^\dagger \qquad (39)$$

where $\mathcal{P}_\sigma$ projects states onto the manifold of visible nodes. The Green function formalism makes clear that, even though all the contributing Hamiltonian are local, two-body in nature, the self energy term in $\mathcal{H}_\sigma^{\mathrm{eff}}$ can contain non-local, many-body interactions encoded in the entanglement Hamiltonian in Eq. 34, as illustrated in Fig. 3, which can be made formally explicit by perturbation or cumulant expansion. Figure 2(c) shows an example of the RBM description for the data obtained from projections on a small Wilson loop of a lattice model.

As we are to discuss in detail in the following sub-section, RBM is a specific implementation of the above picture, where $H_\sigma(\mathbf{a})$ and $H_\tau(\mathbf{b})$ have only one-body energy contribution to the total Hamiltonian with $\mathbf{a}, \mathbf{b}$ coupling parameters; and $\mathcal{H}_{\tau\sigma}(\mathbf{w})$ encodes all two-body interactions $w_{ij}$ between the visible and hidden nodes. By tuning the coupling parameters $\mathbf{a}, \mathbf{b}, \mathbf{w}$, and tracing out the redundant degrees of freedom $\tau$, we arrive at

$$\mathcal{H}^{\text{eff}} = \sum_{s=1}^{N} \sum_{k_1 < \cdots < k_s} \mathcal{I}^{(s)}_{k_1, \cdots, k_s} \sigma_{k_1} \cdots \sigma_{k_s} \quad (40)$$

where $\mathcal{I}^{(s)}_{k_1, \cdots, k_s}$ is a function of $\mathbf{a}, \mathbf{b}, \mathbf{w}$; and it gives the effective coupling between $s$ visible nodes $\sigma_{k_1}, \cdots, \sigma_{k_s}$. Indeed, the ability to encode arbitrarily high order interactions makes RBM a universal representation for all statistical distributions.

In general, a large $n$-body mutual information $I_n$ indicates a large $\mathcal{I}^{(n)}_{k_1, \cdots, k_n}$ in the effective energy of RBM defined by $p(\sigma) \sim \exp(-\mathcal{H}^{\text{eff}})$, which is equivalent to Eq. 34 for the diagonal elements, and will be discussed in detail in the next subsection. A similar idea by the multi-layered deep Boltzmann machine has been used to resolve the non-local entanglement features on the boundary by a local Ising model in the context of holographic duality [60, 61]. One contribution of our work is to make explicit the form of $\mathcal{I}^{(n)}_{k_1, \cdots, k_n}$ of an RBM for $\sigma \in \{+1, -1\}$ relevant for projective samples of spin-$\frac{1}{2}$, and also for other binary bits, without involving perturbation or cumulant expansion, making it easy to extract interaction strength of arbitrary order. This allows us to interrogate the RBM the existence of non-local many-body correlation between sampled degrees of freedoms, where a large $\mathcal{I}^{(n)}$ of the trained network is indicative of $n$-order correlation as TEE encoded in the entanglement Hamiltonian under the basis by which Wilson loop is diagonal. Furthermore, we also present the effective interaction for $\sigma \in \{0, 1\}$ in the Appendix, which can be used to represent high-order density-density interaction in fermion models [26]; however, since any presence of zero would lead to a trivial energy contribution in Eq. 40 regardless of the order, it is not capable of representing non-trivial many-body correlation like Wilson loops of spins.

## B. Large high-order mutual information requires high-order interaction

Previous section has established the possibility of representing the (projected) entanglement Hamiltonian by means of an effective many-body Ising model with arbitrary orders of interactions due to RBM. However, we would like to point out the following caveat that needs clarification: High-order interactions and high-order correlations address fundamentally different aspects of a system: the former address the effective Hamiltonian dynamics that stabilizes low-energy states, while the latter

focus on emergent statistical properties of these states. Indeed, as pointed out in Ref. [1, 4], high-order mutual information $I_{n \geq 3}$ can arise in frustrated Ising models with only pair-wise interactions. For example, consider a simple frustrated Ising model with three binary spins, whose Hamiltonian is given by

$$\mathcal{H}_3 = -\mathcal{I}^{(2)}(\sigma_1\sigma_2 + \sigma_1\sigma_3 + \sigma_2\sigma_3) - \mathcal{I}^{(3)}\sigma_1\sigma_2\sigma_3 \quad (41)$$

This three-spin model is pair-wisely frustrated when $\mathcal{I}^{(3)} = 0$ and $\mathcal{I}^{(2)} < 0$, and straightforward calculation shows the third order mutual information is negative, indicating a frustration-induced irreducible three-body correlation. Therefore, one may ask if an $n$-th order correlation in a closed Wilson loop can also be reflected in lower order interactions $\mathcal{I}^{(n' < n)}$, causing a large degeneracy in RBM representation.

Here we show that such confusion due to the potential degeneracy of representation does not happen in representing the non-local correlation of a TO with $S_{\text{topo}} = -\log 2$. Indeed, even though the frustrated model $\mathcal{H}_3$ gives rise to a negative $I_3$ in absence of high-order interaction $\mathcal{I}^{(3)}$, it cannot generate the high-order mutual information as large as $I_3 = S_{\text{topo}} = -\log 2$ (See Eq. 33) without a dominant $\mathcal{I}^{(3)}$. Straightforward calculation shows

$$\lim_{\mathcal{I}^{(2)} \to -\infty} I_3[\mathcal{H}_3(\mathcal{I}^{(2)}, \mathcal{I}^{(3)} = 0)] = -\log\left(\frac{9}{8}\right) \quad (42)$$

which obviously deviates from $-\log 2$ for a $Z_2$ gauge theory. Therefore, a large value of $\mathcal{I}^{(3)}$ is required in the optimized RBM in order to generate the correct many-body correlation in the projectively measured data. As shown in Fig. 4(a,b), a faithful RBM representation of $S_{\text{topo}} = -\log 2$ is only possible with the highest-order interaction $\mathcal{I}^{(3)}$, and parameters in the effective Hamiltonian of the RBM must flow along the direction where $\mathcal{I}^{(3)}$ increases. The same holds in a four-spin model, as shown in Fig. 4(c,d), which we will demonstrate by our RBM implementation in the following sections. Indeed, this is similar to the classical picture of topological order, where the topological entropy can be perceived as thermally mixed classical loops with energetic constraints [49]. By induction one can show that high-order scenarios have the same properties, which we do not enumerate in this paper.

## C. Interrogate the restricted Boltzmann machine

In this section, we discuss how to interrogate a trained RBM to extract effective interactions of arbitrary order. The physical spins are to be obtained by projective measurements with a set of definite projection basis. The network of RBM is given by the energy function

$$\mathcal{H}(\sigma, \tau) = \sum_i -a_i\sigma_i - b_i\tau_i - \sum_j w_{ij}\sigma_i\tau_j \quad (43)$$

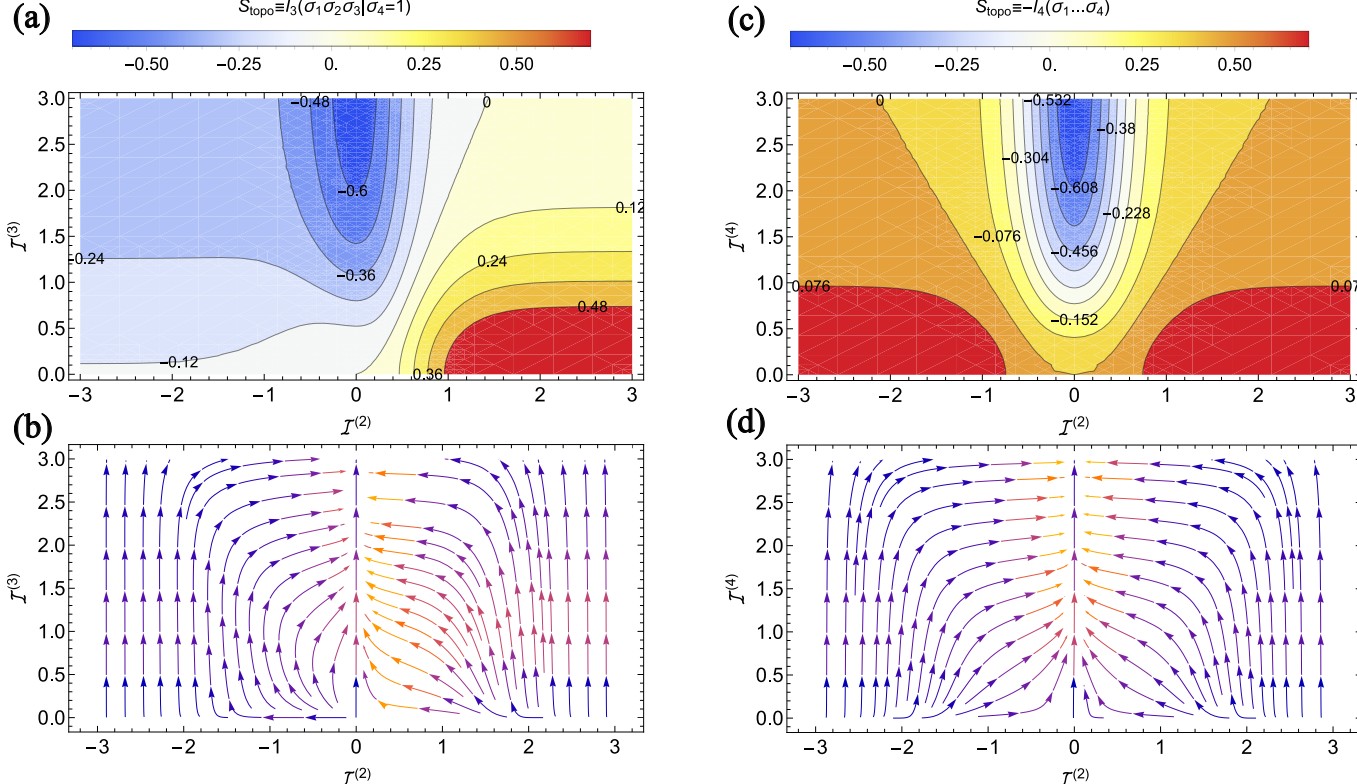

FIG. 4. High-order mutual information as a function of different orders of interaction. (a) The third order mutual information as a function of two- and three-body interactions ($\mathcal{I}^{(2)}$ and $\mathcal{I}^{(3)}$ in Eq. 41). Even though a negative $I_3$ may be present in absence of a three-body interaction $\mathcal{I}^{(3)}$, it requires $|\mathcal{I}^{(3)}| \gg |\mathcal{I}^{(2)}|$ in order to capture the $-\log 2$ topological entropy in the projected data (See Eq. 33). (b) The negative gradient of $I_3$. The arrows denote the optimization direction of effective coupling parameters in an RBM with three-spin input (conditioned on $\sigma_4$). (c) The fourth order mutual information as a function of two- and four-body interactions in the frustrated Hamiltonian $\mathcal{H}_4 = \mathcal{I}^{(2)}(\sigma_1\sigma_2 - \sigma_2\sigma_3 + \sigma_3\sigma_4 + \sigma_1\sigma_4) - \mathcal{I}^{(4)}\sigma_1\sigma_2\sigma_3\sigma_4$. Similar to the former case, it requires $|\mathcal{I}^{(4)}| \gg |\mathcal{I}^{(2)}|$ in order to capture the $-\log 2$ topological entropy in the projected data. (d) The negative gradient of $I_4$. The arrows denote the optimization direction of effective coupling parameters in an RBM with four-spin input.

and the joint probability distribution is $p(\sigma, \tau) = \frac{1}{Z} e^{-\mathcal{H}(\sigma,\tau)}$ with $Z$ the partition function. The probability distribution of the physical spins $\sigma$ can be obtained by marginalization over unphysical spins

$$p(\sigma) = \text{Tr}_\tau \, p(\sigma, \tau) = \frac{e^{-\mathcal{H}(\sigma)}}{Z'} \qquad (44)$$

where the associated partition function is

$$Z' = \frac{Z}{\text{Tr}_\tau \exp(\sum_i b_i \tau_i)} \qquad (45)$$

The the effective energy $\mathcal{H}(\sigma)$ are given by

$$\mathcal{H}(\sigma) = \sum_i a_i \sigma_i + K\left(\sum_i w_{ij}\sigma_i\right) \qquad (46)$$

where $K\left(\sum_i w_{ij}\sigma_i\right)$ is the cumulant generating function

$$K\left(\sum_i w_{ij}\sigma_i\right) = \log \text{Tr}_\tau \left[\exp\left(\sum_{ij} w_{ij}\sigma_i\tau_j\right)\rho(\tau)\right] \qquad (47)$$

and $\rho(\tau)$ is a probability density function of unphysical spins

$$\rho(\tau) = \exp\left(\sum_i b_i \tau_i\right) \Big/ \text{Tr}_\tau \exp\left(\sum_i b_i \tau_i\right) \qquad (48)$$

Here we use the spin states in $\{+1, -1\}$, and we seek to expand the $\mathcal{I}$ in Eq. 40, which usually requires a cumulant expansion of Eq. 47 to arbitrary orders of $\sigma$ such that different orders or correlation become explicit:

$$K(x) = \sum_n \frac{1}{n!} \kappa^{(n)} x^n, \quad x \equiv \sum_i w_{ij}\sigma_i \qquad (49)$$

where $\kappa^{(n)}$ is the $n$th cumulant function

$$\kappa^{(n)} = \frac{\partial^n}{\partial x^n} K(x)\Big|_{x=0} \qquad (50)$$

Indeed, this is usually the method used to extract or construct effective interactions in RBM [26, 41, 62]. However, the $\kappa^{(n)}$ in Eq. 50 requires high order derivative in presence of many-body interaction, making it inconvenient

to analytically track the effective interaction of arbitrary order and evaluate $\mathcal{I}^{(n)}$ for large $n$. Hence, the implementation in previous pioneering works have exploited only the first few orders of interaction.

Here, instead of using conventional cumulant expansion, we propose a projective construction and derive the closed form of effective interaction at arbitrary order without involving derivatives or cumulant functions. For $\sigma \in \{+1, -1\}$ which is anti-symmetric in the $z$ basis of the Pauli matrix, it is intuitively clear that $r$th order interaction should be captured by

$$\mathcal{I}^{(r)}_{k_1,\cdots,k_r} \sim \mathrm{Tr}_{\{\sigma_1,\cdots,\sigma_N\}}\left[\prod_i^r \sigma_{k_i} \mathcal{H}^{\mathrm{eff}}\right], \ \sigma_i = \pm 1 \quad (51)$$

since all energy contributions are cancelled due to the anti-symmetry of $\sigma^z$, except for those of $\sigma_{k_1},\cdots,\sigma_{k_r}$ which are made symmetric by $\prod_i^r \sigma_{k_i}$. Indeed, we will show this representation of $\mathcal{I}^{(r)}$ is true in the case of $\sigma \in \{+1, -1\}$ and is suitable for the application in detecting $I_n$ of emergent gauge field. Yet, in cases such as $\sigma \in \{0, 1\}$ or others, i.e. in absence of anti-symmetry, we need to explicitly construct the effective interaction using a different method. Below we present a general construction that is valid for all binary $\sigma$. The trick is to construct proper resolution of identity which makes all orders of interaction explicit at once. For convenience, we here present the derivation for $\sigma \in \{+1, -1\}$, and use Dirac notation in the $z$ basis of Hilbert space with zero off-diagonal elements. In the appendix we present another example of such construction, where binary spins can be either 0 or 1. We first define the projector $\mathcal{P}_k$ that selects the state where all but the $k$th visible node $\sigma_k$ are $-1$:

$$\mathcal{P}_k = |\sigma_k = 1; \sigma_{k' \neq k} = -1\rangle \langle \sigma_k = 1; \sigma_{k' \neq k} = -1| \quad (52)$$

Then the global projection on $\vec{\sigma}$ supported on the subspace of visible nodes that selects out states where there exists only one visible active node is defined by

$$\mathcal{P}^{(1)} \equiv \sum_{k=1}^N \mathcal{P}_k \quad (53)$$

Similarly for any positive integer $m \leq N$ we can define the projector $\mathcal{P}^{(m)}$ that selects $m$ out of $N$ visible nodes that are active:

$$\mathcal{P}^{(m)} = \sum_{k_1 < \cdots < k_m} \mathcal{P}_{k_1,\cdots,k_m} \quad (54)$$

where $\mathcal{P}_{k_1,\cdots,k_m}$ is the projector which selects the state with $\sigma_{k \in \{k_1,\cdots,k_m\}} = 1$ and others $-1$:

$$\mathcal{P}_{k_1,\cdots,k_m} = \left|\sigma_{k \in \{k_1,\cdots,k_m\}} = 1; \sigma_{k' \notin \{k_1,\cdots,k_m\}} = -1\right\rangle \\ \left\langle\sigma_{k \in \{k_1,\cdots,k_m\}} = 1; \sigma_{k \notin \{k_1,\cdots,k_m\}} = -1\right| \quad (55)$$

These give a useful resolution of identity

$$\sum_{m=0}^N \mathcal{P}^{(m)} = I \quad (56)$$

by which the Eq. 47 as a diagonal operator can be readily written as combination of different groups:

$$\hat{K}(\mathbf{w}^\intercal \vec{\sigma}) = \sum_{\vec{\sigma}} \sum_{m=0}^N \sum_{k_1 < \cdots < k_m} K_m(\mathbf{w}) \\ \times \delta_{\sigma_{k \in \{k_1,\cdots,k_m\}},1} \ \delta_{\sigma_{k \notin \{k_1,\cdots,k_m\}},-1} \\ \times \left|\sigma_{k \in \{k_1,\cdots,k_m\}} = 1; \sigma_{k' \notin \{k_1,\cdots,k_m\}} = -1\right\rangle \langle \vec{\sigma}| \quad (57)$$

where for convenience we have defined

$$K_m(\mathbf{w}^\intercal_{k_j}) \equiv K\left(\sum_{j=1}^m \mathbf{w}^\intercal_{k_j} - \sum_{j=m+1}^N \mathbf{w}^\intercal_{k_j}\right) \quad (58)$$

Noting that $\sigma_k$ is binary and classical, we write the Kronecker delta in Eq. 57 as

$$\delta_{\sigma_{k \in \{k_1,\cdots,k_m\}},1} \ \delta_{\sigma_{k \notin \{k_1,\cdots,k_m\}},-1} \\ = \prod_{i=1}^m \left(\frac{1+\sigma_{k_i}}{2}\right) \prod_{k \notin \{k_1,\cdots,k_m\}} \left(\frac{1-\sigma_k}{2}\right) \quad (59)$$

Hence the diagonal terms of $\hat{K}$ given by the trace over $\sigma$ is

$$K(\mathbf{w}^\intercal \vec{\sigma}) = \sum_{m=0}^N \sum_{k_1 < \cdots < k_m} K_m(\mathbf{w}^\intercal_{k_j}) \\ \times \prod_{i=1}^m \left(\frac{1+\sigma_{k_i}}{2}\right) \prod_{k \notin \{k_1,\cdots,k_m\}} \left(\frac{1-\sigma_k}{2}\right) \quad (60)$$

where we can expand the two products respectively. The first product in Eq. 60 is then written into:

$$\prod_{i=1}^m \left(\frac{1+\sigma_{k_i}}{2}\right) = \frac{1}{2^m} \sum_{q=0}^m \sum_{k_1 < \cdots < k_q} \sigma_{k_1} \cdots \sigma_{k_q} \quad (61)$$

and the other product into:

$$\prod_{k \notin \{k_1,\cdots,k_m\}} \left(\frac{1-\sigma_k}{2}\right) = \frac{1}{2^{N-m}} \sum_{p=0}^{N-m} (-1)^p \\ \times \left(\sum_{k_{m+1} < \cdots < k_{m+p}} \sigma_{k_{m+1}} \cdots \sigma_{k_{m+p}}\right) \quad (62)$$

Then, combining them together we have the $K$ in the following form:

$$K(\mathbf{w}^\intercal \vec{\sigma}) = \sum_{m=0}^N \sum_{p=0}^{N-m} \sum_{\substack{k_1 < \cdots < k_m; \\ k_{m+1} < \cdots < k_{m+p} \\ \neq (k_1,\cdots,k_m)}} K_m(\mathbf{w}^\intercal_{k_j}) \\ (-1)^p \left[\sum_{q=0}^m \left(\sum_{k_1 < \cdots < k_q} \sigma_{k_1} \cdots \sigma_{k_q}\right) \sigma_{k_{m+1}} \cdots \sigma_{k_{m+p}}\right] \quad (63)$$

where we have left out the constant prefactor $\frac{1}{2^N}$. Given the size of sample space $N$, each term in the above expression will be determined by a triplet $(m, q, p)$ where $q$ is upper bounded by $m$ and $p$ by $N - m$. This looks complicated and hard to rearrange. In order to write down an explicit energy function for $r$-th order of interaction, the above summation can be grouped according to different doublet $(p, q)$, which determines the order $r = p + q$. we replace the index $q$ by $q = r - p$, we have

$$K^{(r)}(\mathbf{w}^\intercal \vec{\sigma}) = \sum_{m=0}^{N} \sum_{p=0}^{N-m} \sum_{\substack{k_1 < \cdots < k_m; \\ k_{m+1} < \cdots < k_{m+p} \\ \neq (k_1, \cdots, k_m)}} K_m(\mathbf{w}^\intercal_{k_j})$$

$$(-1)^p \left[ \sum_{\substack{k_1 < \cdots < k_{r-p}; \\ k_i \in \{k_1, \cdots, k_m\}}} \sigma_{k_1} \cdots \sigma_{k_{r-p}} \sigma_{k_{m+1}} \cdots \sigma_{k_{m+p}} \right] \tag{64}$$

where the first $r - p$ spins $\sigma_{k_1} \cdots \sigma_{k_{r-p}}$ in the product are attributed to the projection into $\sigma = +1$, i.e. from Eq. 61, thus are responsible for $+\mathbf{w}^\intercal_{k_{s_i}}$ in $K$, where the nested label $k_{s_i}$ ($1 \leq i \leq r - p$) is for the $\sigma_{k_{j_s}}$ – the $r - p$ out of $r$ spins that are chosen to be projected into $+1$; the spins $\sigma_{k_{m+1}} \cdots \sigma_{k_{m+p}}$ in the product are attributed to the projection into $\sigma = -1$, i.e. Eq. 62, thus are responsible for $-\mathbf{w}^\intercal_{k_{j_i}}$ in $K$, and we group these spins using labels $k_{j_i}$ with $p = |\{k_{j_i}\}|$; finally the rest $N - r$ spins, though not in the $r$ spins of interest, still contribute to energy (in contrast to the case of $\sigma \in \{1, 0\}$) and are associated with $+\mathbf{w}^\intercal_{k_{l_i}}$ for $1 \leq i \leq N - r$. Therefore, by rearranging indices it is straightforward to read out the interaction strength $\mathcal{I}^{(r)}_{k_1, \cdots, k_r}$ between $r$ spins $\sigma_{k_1} \cdots \sigma_{k_r}$:

$$\mathcal{I}^{(r)}_{k_1, \cdots, k_r} = \sum_{\substack{\{k_{j_i}\}, \{k_{s_i}\} \\ \subseteq \{k_1, \cdots, k_r\}}} \sum_\eta (-1)^p K\left( - \sum_{i=1}^{p} \mathbf{w}^\intercal_{k_{j_i}} \right.$$
$$\left. + \sum_{i=1}^{r-p} \mathbf{w}^\intercal_{k_{s_i}} + \sum_{i=1}^{N-r} \eta^{k_{l_i}} \mathbf{w}^\intercal_{k_{l_i}} \right) \tag{65}$$

where in indeces are grouped according to

$$\{k_{j_i}\} \cap \{k_{s_i}\} = \emptyset, \tag{66}$$
$$\{k_{j_i}\} \cup \{k_{s_i}\} = \{k_1, \cdots, k_r\}, \tag{67}$$
$$k_{l_i} \in \{k_{r+1}, \cdots, k_N\} \tag{68}$$

and $\sum_\eta$ is the summation over all vectors $\eta$ of dimension $|\{k_{j_l}\}| = N - r$ whose elements $\eta^{k_{j_l}}$ are $\pm 1$ binaries. It is then easy to see the compact equivalent form of Eq. 65:

$$\mathcal{I}^{(r)}_{k_1, \cdots, k_r} = \text{Tr}_{\{\sigma_1, \cdots \sigma_N\}} \left[ \sigma_{k_1} \cdots \sigma_{k_r} K\left( \sum_i w_{ij} \sigma_i \right) \right] \tag{69}$$

which is consistent with Eq. 51. This is our central result for the RBM representation. We will test the strength

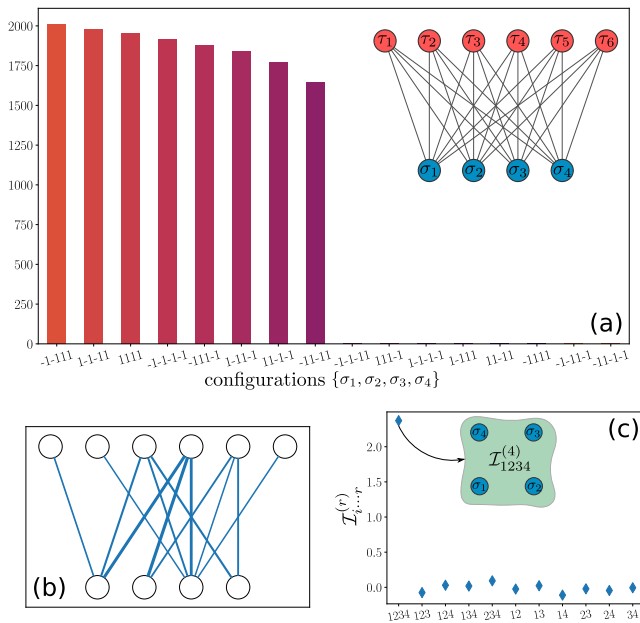

FIG. 5. (a) The number of samples corresponding to different spin configurations from the trained RBM. The eight dominant configurations have zero flux in accordance with the gauge constraint of TC. The inset shows the network structure of the RBM with six hidden spins and four visible spins samples from a plaquette loop of TC. (b) Visualization of the weight matrix $\mathbf{w}$ of the trained RBM. Thick and thin blue lines indicates strong and weak coupling. (c) The magnitude of effective interactions of different orders. The strongest interaction is of the fourth order corresponding to the inset figure.

of the formalism in the coming section under the context of $Z_2$ TO such as toric code that harbors non-local, many-body correlation in reduced density matrices. The construction for $\sigma \in \{0, 1\}$ is presented in Appendix. A for other potential applications.

## V. RBM REPRESENTATION OF THE HIGH-ORDER CORRELATION IN TC

The smallest unit that exhibits topological entanglement in TC is the four-point plaquette operator, which is also the smallest Wilson loop that is gauge invariant in the vortex-free ground state. As discussed in previous sections, the topological entanglement in this minimal case can be understood as higher order mutual information $I_4$ that is encoded in the local density matrix. In this section, we use RBM network to represent the reduced density matrix of TC in the basis whereby the Wilson loop is explicit, i.e. $\prod_{\Box_i} \sigma_i^z$ or $\prod_{+_i} \sigma_i^x$; and apply the previously derived result to capture the four-body correlation by the effective four-body interaction of the trained RBM.

We first disentangle Eq. 65 or Eq. 69 and write down explicitly its first few orders of interaction, up to fourth order by which a Wilson loop is built. In this simple

exemplary case $N = 4$, hence, each order of effective interaction can be represented by the following way

$$
\begin{aligned}
\mathcal{I}_{1,2}^{(2)} = \sum_{\eta} & K\left(-\mathbf{w}_1^\mathsf{T} - \mathbf{w}_2^\mathsf{T} + \eta^1 \mathbf{w}_3^\mathsf{T} + \eta^2 \mathbf{w}_4^\mathsf{T}\right) \\
& + K\left(\mathbf{w}_1^\mathsf{T} + \mathbf{w}_2^\mathsf{T} + \eta^1 \mathbf{w}_3^\mathsf{T} + \eta^2 \mathbf{w}_4^\mathsf{T}\right) \\
& - K\left(\mathbf{w}_1^\mathsf{T} - \mathbf{w}_2^\mathsf{T} + \eta^1 \mathbf{w}_3^\mathsf{T} + \eta^2 \mathbf{w}_4^\mathsf{T}\right) \\
& - K\left(-\mathbf{w}_1^\mathsf{T} + \mathbf{w}_2^\mathsf{T} + \eta^1 \mathbf{w}_3^\mathsf{T} + \eta^2 \mathbf{w}_4^\mathsf{T}\right)
\end{aligned}
\tag{70}
$$

and so on, where $\eta^i = \pm 1$. Note that the sign of the prefactor in each summand is determined by the number of minus signs of the first twospins, i.e. the $(-1)^p$ of Eq. 65, whereby even (odd) number of minuses of the first twospins gives the positive (negative) prefactor. The third order $\mathcal{I}^{(3)}$ and fourth order $\mathcal{I}^{(4)}$ can be expressed by the same token, with eight and sixteen summands respectively.

Results of the RBM are presented in Fig. 5. The network structure is shown in the inset of Fig. 5(a), where we used six nodes in the hidden layer to capture the joint probability distribution of 5000 projective samples taken under $z$ basis in the four spins in a plaquette. Note that it takes at least $n$ hidden nodes in order to represent an $n$th order effective interaction between the visible spins. The interaction matrix $\mathbf{w}$ of the trained RBM is showcased in Fig. 5(b), where thinker lines indicate stronger coupling between visible and hidden spins. These together give the effective fourth order interaction, while leaving lower orders of effective interaction negligible. The comparison between different orders of interaction are shown in Fig. 5(c), where, as expected from the fact that the four spins are entangled collectively, the fourth order effective interaction between the $\sigma_1 \sigma_2 \sigma_3 \sigma_4$ is significantly larger than other lower order interactions. The validity of the model is further verified by the direct sampling from the RBM network, as shown in Fig. 5(a), the first eight configurations whose product equal to $+1$ are dominant in frequency over those whose product are $-1$. The same method would work equally well in other topologically ordered system with a richer Hilbert space, such as the Kitaev spin liquid, the paradigmatic integrable TO model defined on the honeycomb lattice. Its eigen function factorizes into the gauge sector and majorana fermion sector $|\Psi\rangle = \sum_{\mathcal{G}} |M_{\mathcal{G}}\rangle \otimes |\mathcal{G}\rangle$, of which only the gauge sector $|\mathcal{G}\rangle$ contribute to the topological entanglement entropy of the Wilson loop of spins [23, 63]. The smallest Wilson loop in Kitaev model is the six-point hexagon with alternating spin basis, which requires at least 6 hidden nodes to represent the sixth order correlation in the projective samples. The logic in the TEE of the Kitaev model is the same as that presented for TC model, so we do not repeat the sampling thereof.

In this minimal example we have presented the witnessing of TEE in TC using the projective samples under basis that is proper to the Wilson loop. Even under weak perturbation, we expect that the projective samples in this basis would exhibit the same dominant statistical correlation. Furthermore, under random basis measurement, the statistical correlation relevant for TEE may not be present under a particular chosen basis combination. However, enabled by the low computational cost, it is always doable to train multiple RBMs and extract the effective interaction for each of them; and the TEE by would be reflected by the existence of a high-order effective interaction, which also directly inform the form of Wilson loop explicitly.

## VI. CONCLUSION AND OUTLOOK

In this work we propose a statistical interpretation which unifies the high-order correlation and topological entanglement under the same statistical framework. We demonstrate in Sec. II that the existence of a non-zero TEE can be understood in the statistical view as the emergent $n$th order mutual information $I_n$ (for arbitrary $n \geq 3$) reflected in projectively measured samples, which also makes explicit the equivalence between the two existing methods for its extraction – the Kitaev-Preskill and the Levin-Wen construction. The statistical nature of $I_n$ can be reflected in the effective $n$th order mutual information, as is discussed in a minimal example in Sec. III B. Hence, by exploiting the universal representational power of RBM, $I_n$ can be described by the effective interaction between visible nodes of a trained RBM as a descriptor of the distribution of quantum sampling of spins. In Sec. V, we explicitly showcased the construction of the RBM which captures the high-order correlation and/or topological entanglement that are encoded in the distribution of projected sample. Furthermore, in order to extract the coefficient of each order of interaction, we developed in Sec. IV C a method to interrogate the trained RBM, making explicit the analytical form of arbitrary order of interaction relevant for $I_n$ in terms of the effective Hamiltonian $\mathcal{H}$, whereby the high-order correlation is reflected in the effective many-body interaction between visible nodes after tracing out the hidden nodes. Recently the topological phase of TC is realized in cold atom setup [16, 17]. Through our statistical perspective and concrete neural network construction, we hope to provide useful insights in these relevant investigation of topologically ordered matter.

Beyond faithfully describing the high-order correlation, such exact extraction of the effective interactions up to arbitrary order opens the door for various application in many-body physics. Indeed, the effective many-body interaction encoded by the RBM network has been exploited in many-body systems. For example, in [41], where authors successfully used an RBM to capture interaction matrix of an Ising model; and [26] where the authors used RBM to exactly represent the interaction between nucleons. In our work, the exact extraction of $n$th order interaction presented in Eq. 65 allows us to step further into arbitrarily high order of interactions, and a generic

Hubbard-Stratonovich-type transformation, i.e. the representation of many-body interacting Hamiltonians in terms of two-body Hamiltonians with auxiliary fields. A potential application is a many-nucleon interaction:

$$H^V_{ij} = v_{ij}\hat{n}_i\hat{n}_j + v_{ijk}\hat{n}_i\hat{n}_j\hat{n}_k + v_{ijkl}\hat{n}_i\hat{n}_j\hat{n}_k\hat{n}_l + \cdots \quad (71)$$

where $v$ are scalar coefficients dependent on nucleon coordinates, and a functional of spin and isospin. It is then possible to introduce an auxiliary field so as to provide a simpler representation with fewer orders of interaction. Indeed, this is carried out in Ref. [26] where authors used the hidden nodes of RBM as an auxiliary field $h$ to disentangle the third order interaction into two-body interactions between $h$ and $n$. With the representation of high-order interaction, we hope inspire auxiliary field construction that decouples arbitrarily high-order interactions between nucleons and other many-body interacting systems.

## ACKNOWLEDGMENTS

S. Feng acknowledges support from NSF Materials Research Science and Engineering Center (MRSEC) Grant No. DMR-2011876 and the Presidential Fellowship of The Ohio State University. N. Trivedi acknowledges support from NSF-DMR 2138905. D. Kong acknowledges support from UCLA Graduate Fellowship. We thank Xiaozhou Feng for insightful discussions on RBM. We also thank Kevin Zhang, Yuri Lensky and Eun-Ah Kim for enlightening discussions during our collaboration on supervised machine learning.

## Appendix A: Representation for $\sigma \in \{0, 1\}$

The logic is the same as the that for $\sigma \in \pm 1$, except that the representation of projection operator is changed. In the $\{0, 1\}$ case, we apply the following resolution of identity for any positive integer $m \le N$:

$$\mathcal{P}^{(m)} = \sum_{k_1 < \cdots < k_m} \mathcal{P}_{k_1 \cdots k_m}, \quad (A1)$$

where $\mathcal{P}^{(m)}$ is defined as

$$\mathcal{P}_{k_1,\cdots,k_m} = \big|\sigma_{k\in\{k_1,\cdots,k_m\}} = 1; \sigma_{k'\notin\{k_1,\cdots,k_m\}} = 0\big\rangle \\ \big\langle \sigma_{k\in\{k_1,\cdots,k_m\}} = 1; \sigma_{k\notin\{k_1,\cdots,k_m\}} = 0\big| \quad (A2)$$

and the completeness is then given bycc $\sum_{m=0}^{N} \mathcal{P}^{(m)} = I$. In the diagonal case, it is straightforward to write the cumulant generating function in the matrix form:

$$\hat{K}(\mathbf{w}^\mathsf{T}\sigma) = \sum_\sigma \log\left[\sum_\tau \exp(\sigma\mathbf{w}^\mathsf{T}\tau)\rho(\tau)\right]|\sigma\rangle\langle\sigma| \quad (A3)$$

Attach to it the resolution of identity in terms of projectors:

$$\hat{K}(\mathbf{w}^\mathsf{T}\hat{\sigma}) = \sum_{m=0}^{N}\sum_\sigma \sum_{k_1<\cdots<k_m} K\left(\sum_{j=1}^{m}\mathbf{w}^\mathsf{T}_{k_j}\right) \\ \times \delta_{\sigma_{k\in\{k_1,\cdots,k_m\}},1}\ \delta_{\sigma_{k\notin\{k_1,\cdots,k_m\}},0} \\ \times \big|\sigma_{k\in\{k_1,\cdots,k_m\}} = 1; \sigma_{k'\notin\{k_1,\cdots,k_m\}} = 0\big\rangle\langle\sigma| \quad (A4)$$

In the classical case whereby $\sigma_k$ is binary, we write the Kronecker delta as

$$\delta_{\sigma_{k\in\{k_1,\cdots,k_m\}},1}\ \delta_{\sigma_{k\notin\{k_1,\cdots,k_m\}},0} = \prod_{i=1}^{m}\sigma_{k_i}\prod_{k\notin\{k_1,\cdots,k_m\}}(1-\sigma_k) \quad (A5)$$

Hence the diagonal terms of $\hat{K}$ given by the trace over $\sigma$ is

$$K(\mathbf{w}^\mathsf{T}\sigma) = \sum_{m=0}^{N}\sum_{k_1<\cdots<k_m} K\left(\sum_{j=1}^{m}\mathbf{w}^\mathsf{T}_{k_j}\right) \\ \times \prod_{i=1}^{m}\sigma_{k_i}\prod_{k\notin\{k_1,\cdots,k_m\}}(1-\sigma_k) \quad (A6)$$

we expand the last term into

$$\prod_{k\notin\{k_1,\cdots,k_m\}}(1-\sigma_k) = \sum_{p=0}^{N-m}(-1)^p \\ \times \left(\sum_{k_{m+1}<\cdots<k_{m+p}}\sigma_{k_{m+1}}\cdots\sigma_{k_{m+p}}\right) \quad (A7)$$

Then we have the effective interaction for $s$th order:

$$K(\mathbf{w}^\mathsf{T}\sigma) = \sum_{m=0}^{N}\sum_{p=0}^{N-m}\sum_{k_1<\cdots<k_m}(-1)^p K\left(\sum_{j=1}^{m}\mathbf{w}^\mathsf{T}_{k_j}\right) \\ \times \sigma_{k_1}\sigma_{k_2}\cdots\sigma_{k_{m+p}} \quad (A8)$$

This is equivalent to the result derived in Ref. [64] for $\sigma \in \{0, 1\}$ using a different method. By rearranging the induces, it is straightforward to get

$$\mathcal{I}^{(s)}_{k_1,\cdots,k_s} = \sum_{p=0}^{s-1}(-1)^p\sum_{j_1<\cdots<j_{s-p}}^{s} K\left(\sum_{i=1}^{s-p}\mathbf{w}^\mathsf{T}_{k_{j_i}}\right) \quad (A9)$$

This result can be used to represent high-order density-density interaction in fermion models [26]; however, since any presence of zero would lead to a trivial energy contribution in the effective energy regardless of the order of interaction, hence, it is not capable of representing non-trivial many-body correlation/interaction of spins, which requires the representation for $\sigma \in \{+1, -1\}$ as discussed in the main text.

## Appendix B: Training of Restricted Boltzmann machine

An RBM is a bipartite binary probabilistic graphical model corresponding to the following distribution,

$$p(\sigma, \tau) = \frac{1}{Z} \exp[-E(\sigma, \tau)] \tag{B1}$$

which assigns a probability to every possible pair of a visible ($\sigma$) and a hidden vector ($\tau$) via this energy function function:

$$E(\sigma, \tau) = -\sum_{i \in \text{visible}} a_i \sigma_i - \sum_{j \in \text{hidden}} b_j \tau_j - \sum_{i,j} w_{ij} \sigma_i \tau_j \tag{B2}$$

The probability of $\sigma$ or $\tau$ is given by a marginalization:

$$p(\sigma) = \frac{1}{Z} \sum_\tau \exp[-E(\sigma, \tau)], \;\; p(\tau) = \frac{1}{Z} \sum_\sigma \exp[-E(\sigma, \tau)] \tag{B3}$$

The derivation of the log probability w.r.t. $w_{ij}$ is:

$$
\begin{aligned}
\frac{\partial \log p(\sigma)}{\partial w_{ij}} &= \frac{1}{p(\sigma)} \left( -\frac{1}{Z^2} \frac{\partial Z}{\partial w_{ij}} \right) \sum_\tau e^{-E(\sigma, \tau)} \\
&+ \frac{1}{p(\sigma)} \sum_\tau \sigma_i \tau_j \frac{e^{-E(\sigma, \tau)}}{Z} \\
&= -\frac{1}{p(\sigma)} \left( \sum_{\tau, \sigma} \sigma_i \tau_j \frac{e^{-E(\sigma, \tau)}}{Z} \right) \left( \sum_\tau \frac{e^{-E(\sigma, \tau)}}{Z} \right) \\
&+ \sum_\tau \sigma_i \tau_j \frac{p(\sigma, \tau)}{p(\sigma)} \\
&= -\sum_{\tau, \sigma} \sigma_i \tau_j p(\sigma, \tau) + \sum_\tau \sigma_i \tau_j p(\tau | \sigma) \\
&= -\mathbb{E}_{\text{model}} [\sigma_i \tau_j] + \mathbb{E}_{\text{data}} [\sigma_i \tau_j]
\end{aligned}
\tag{B4}
$$

this leads to the gradient ascent learning rule of $w_{ij}$:

$$\delta w_{ij} = \beta(\mathbb{E}_{\text{data}} [\sigma_i \tau_j] - \mathbb{E}_{\text{model}} [\sigma_i \tau_j]) \tag{B5}$$

and by the same token we can derive the updating process for $a_i$ and $b_j$:

$$
\begin{aligned}
\delta a_i &= \beta(\mathbb{E}_{\text{data}} [\sigma_i] - \mathbb{E}_{\text{model}} [\sigma_i]) \\
\delta b_j &= \beta(\mathbb{E}_{\text{data}} [\tau_j] - \mathbb{E}_{\text{model}} [\tau_j])
\end{aligned}
\tag{B6}
$$

where $\beta$ is the learning rate.

Now we need to figure out how to calculate the relevant expectation values mentioned above. We start with the conditional expectation $\mathbb{E}_{\text{data}} [\sigma_i \tau_j]$. The key is to sample the probability $p(\tau | \sigma)$. We can easily write down the conditional probability:

$$p(\tau | \sigma) = \frac{p(\tau, \sigma)}{p(\sigma)} = \frac{\frac{1}{Z} e^{-E(\sigma, \tau)}}{\frac{1}{Z} \sum_\tau e^{-E(\sigma, \tau)}} = \frac{e^{-E(\sigma, \tau)}}{\sum_\tau e^{-E(\sigma, \tau)}} \tag{B7}$$

and conditional probability for a single hidden node $\tau_j$ can be derived by marginalization:

$$p(\tau_j | \sigma) = \sum_{\{\tau_k\} - \tau_j} p(\{\tau_k\} | \sigma) = \frac{\sum_{\{\tau_k\} - \tau_j} e^{-E(\sigma, \tau)}}{\sum_\tau e^{-E(\sigma, \tau)}} \tag{B8}$$

For convenience we rewrite the energy function in the following form which separates the hidden and the visible nodes:

$$
\begin{aligned}
E(\sigma, \tau) &= -\sum_{j \in \text{hidden}} \left[ \tau_j \left( b_j + \sum_{i \in \text{visible}} w_{ij} \sigma_i \right) \right] \\
&\quad - \sum_{i \in \text{visible}} a_i \sigma_i \\
&\equiv -\sum_j \gamma_j(\sigma) \tau_j - \sum_i a_i \sigma_i
\end{aligned}
\tag{B9}
$$

so the Boltzmann factor in the numerator now takes the form:

$$\exp[-E(\sigma, \tau)] = \prod_i e^{-a_i \sigma_i} \prod_j e^{-\gamma_j(\sigma) \tau_j} \tag{B10}$$

Therefore the denominator in Eq. B8 can be written as

$$
\begin{aligned}
\sum_\tau e^{-E(\sigma, \tau)} &= \prod_i e^{-a_i \sigma_i} \sum_\tau \prod_k e^{-\gamma_k(\sigma) \tau_k} \\
&= \left[ \prod_i e^{-a_i \sigma_i} \right] \left[ \sum_{\tau_j = \{0,1\}} e^{-\gamma_j(\sigma) \tau_j} \right] \\
&\quad \times \left[ \sum_{\{\tau_k\} - \tau_j} \prod_{k \neq j} e^{-\gamma_k(\sigma) \tau_k} \right]
\end{aligned}
\tag{B11}
$$

and the numerator:

$$
\begin{aligned}
\sum_{\{\tau_k\} - \tau_j} e^{-E(\sigma, \tau)} &= e^{-\gamma_j(\sigma) \tau_j} \left[ \prod_i e^{-a_i \sigma_i} \right] \\
&\quad \times \left[ \sum_{\{\tau_k\} - \tau_j} \prod_{k \neq j} e^{-\gamma_k(\sigma) \tau_k} \right]
\end{aligned}
\tag{B12}
$$

hence Eq. B8 becomes a Logistic form:

$$p(\tau_j | \sigma) = \frac{e^{-\gamma_j(\sigma) \tau_j}}{1 + e^{-\gamma_j(\sigma)}} \tag{B13}$$

Since each element in $\tau_j$ is binary, we can readily write down the conditional probability for $\tau_j = 1, -1$ conditioned on $\sigma$:

$$
\begin{aligned}
p(\tau_j = 1 | \sigma) &= \frac{\exp(-b_j - \sum_i w_{ij} \sigma_i)}{1 + \exp(-b_j - \sum_i w_{ij} \sigma_i)} \\
&= \text{sigmoid} \left( b_j + \sum_i w_{ij} \sigma_i \right)
\end{aligned}
\tag{B14}
$$

$$
\begin{aligned}
p(\tau_j = 0 | \sigma) &= 1 - p(\tau_j = 1 | \sigma) \\
&= \frac{1}{1 + \exp(-b_j - \sum_i w_{ij} \sigma_i)}
\end{aligned}
\tag{B15}
$$

By the same token, we can show $p(\sigma_i|\tau)$, which, however, is no longer a sigmoid function in the case $\sigma \in \{+1, -1\}$. We are now prepared to sample calculate $\mathbb{E}_{\text{data}}[\sigma_i\tau_j] = \sum_\tau \sigma_i\tau_j p(\tau|\sigma)$ for every pair of $i$ and $j$.

---

**Algorithm 1** Sampling $\mathbb{E}_{\text{data}}[\sigma_i\tau_j]$

---

**Input:** Data batch $(\sigma_1, \cdots, \sigma_N)$ and initial parameters of RBM
**Output:** $\mathbb{E}_{\text{data}}[\sigma_i\tau_j]$
1. Initialize the $\mathbf{M} = 0$ matrix
2. For each $\sigma_t$ in data batch:
      Sample $\tau \sim p(\tau|\sigma_t) = \sigma(\mathbf{b} + \mathbf{w}^\top\sigma)$
      $\mathbf{M} \leftarrow \mathbf{M} + \sigma_t\tau^\top$
3. $\mathbb{E}_{\text{data}}[\sigma\tau^\top] \leftarrow \mathbf{M}/N$

---

Next we need to compute $\mathbb{E}_{\text{model}}[\sigma_i\tau_j] = \sum_{\sigma,\tau} \sigma_i\tau_j$, which is significantly harder since we are drawing correlated samples. Nevertheless, note that elements in $\sigma$ or $\tau$ are not correlated within the same layer, so, assuming

convergence is achievable, we can write down a similar algorithm sampling the hidden and visible layer one after another:

---

**Algorithm 2** Sampling $\mathbb{E}_{\text{model}}[\sigma_i\tau_j]$

---

**Input:** Initial parameters of RBM
**Output:** $\mathbb{E}_{\text{model}}[\sigma_i\tau_j]$
1. Initialize the $\mathbf{M} = 0$ matrix
2. Initialize $\sigma$ to be a random vector
3. Repeat $N_c$ times (until convergence):
      Sample $\tau \sim p(\tau|\sigma) = \sigma(\mathbf{b} + \mathbf{w}^\top\sigma)$
      Sample $\sigma \sim p(\sigma|\tau) = \sigma(\mathbf{a} + \mathbf{w}\tau)$
      $\mathbf{M} \leftarrow \mathbf{M} + \sigma\tau^\top$
4. $\mathbb{E}_{\text{model}}[\sigma\tau^\top] \leftarrow \mathbf{M}/N_c$

---

However, this scheme usually converges very slowly since samples of $\tau$ and $\sigma$ are correlated. This is exactly where the contrastive divergence (CD) has a part to play. This can simply be done by setting $N_c = n$ for $\text{CD}_n$, where $n$ is commonly chosen to be $n = 1$.

---

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
