# Peer review of "A statistical approach to topological entanglement: Boltzmann machine representation of high-order irreducible correlations"

_SciPost Physics_

## Round 1 · Referee Report · Anonymous (Referee 1) · 2024-1-9

Strengths

The paper seems to make an interesting link between quantum systems and topological entanglement.

Weaknesses

The part on the RBM is not very convincing, in particular: * the Fig. 4 is quick unclear to me * the expression Eq. 69 is only a slight generalization of what has been done in Ref 64 * the numerical experiments are not well-described * the numerical experiments consider very small system size with a very specific setting * the training procedure described in the appendix is known to be problematic

Report

The work "A statistical approach to topological entanglement: Boltzmann machine representation of high-order irreducible correlations" considers an interpretation of the high-order correlations in quantum system in term of topological entanglement in topologically ordered states of matter at zero temperature. They then construct a Restricted Boltzmann Machine (RBM) to capture the potential higher-order interaction in classical spin systems.

First of all, I would like to make it clear that I am neither an expert on quantum systems nor on topological entanglement. Therefore, I will focus my comments and criticisms mainly on the second part of the paper, which deals with the RBM and its training process and the procedure for extracting higher order interactions.

In section IV I'm not sure if the quantity E has been defined, does it correspond to the potential of the variables?
Also, I find it rather unintuitive to define the effective Hamiltonian, Eq. 40, before the Hamiltonian of the RBM is even mentioned (it only comes in Eqs. 43-46).
I understand that the authors in IV-B stress the importance of distinguishing between higher-order correlations that arise, for instance because of pairwise interactions, or from true higher-order interactions. Then they claim that the correlations due to a Wilson loop can only be represented by higher-order interactions in the Hamiltonian. Is this a general assessment and is there a general way to understand when higher-order interactions are necessary or not? The connection with the non-local correlation of a TO is a bit puzzling to me. All of this seems to be related to Fig. 4, which admittedly remains unclear to me.
It would be much clearer if the authors could rewrite this part and clearly indicate in what range I_3 and I_4 are sufficiently relevant to imply the presence of higher order interactions. Finally, if we consider the gradient (lower part of Fig. 4), this means that all trajectories in this case converge to I_3,I_4 != 0, while I_2 = 0?

In Section IV-C, the formula for the effective interaction is derived by considering the effective Hamiltonian of the RBM on the visible nodes. Most of the derivation is based on previous calculations performed for the case of visible {0,1} variables, while here the authors consider the case of {-1,+1} variables. Although this distinction is quite important for the effective model as pointed out in appendix (page 14, second column, last paragraph), I have to mention that this is a quite straight-forward generalization of the calculation done in ref. [64], not to mention that this reference is only cited in the appendix and not in the main text. For this reason, I do not find sentences like "we developed in Sec. IV C a method to interrogate the trained RBM, making explicit the analytical form of arbitrary order of interaction relevant for In in terms of the effective Hamiltonian H" (in the conclusion) appropriate.
Also, there seems to be a problem with Eq. 69: The expression depends on a hidden node subscript "j". I think the entire expression should be summed over the hidden nodes, as in [64] equation 14.

Section V describes the numerical experiment that confirms the theoretical results and demonstrates the possibility of effectively deriving higher-order interactions. There are several problems here. The most important one is that it is not clearly described how the training set is generated. Since there are only 4 visible nodes, the total number of configurations should be 2^4 = 16, and I don't understand where the number "5000" samples comes from. We also don't know from which model these variables were generated. Finally, the authors state that they use the contrastive divergence CDn for training the RBM (in the appendix). While I agree that the system considered here is so small that sampling the variables is quite easy, it is incredible to still read in the literature that contrastive divergence is a valid procedure for learning the (Restricted)- Boltzmann Machine. In Neurips 2021 "Equilibrium and non-equilibrium regimes in the learning of restricted Boltzmann machines" by Decelle et al. it was shown that this type of method can be quite dangerous for the learning and that it is necessary to monitor the relaxation time of the RBM, and in Neurips 2019 "Learning non-convergent non-persistent short-run mcmc toward energy-based mode" by Nijkamp et al. the same effect in energybased models.

Finally, The numerical results are quite simple. The authors do some tests with an RBM consisting of 4 visible and 6 hidden nodes. Given such a small size, it is not clear to me whether their method demonstrates the feasibility of their approach. The first problem is that the analytical expression Eq. 69 is also not applicable when the number of visible nodes is greater than ~25 (because of the trace over all spins). The second problem is that the test is only performed for a system containing only an interaction of order 4 without any pairwise term.
Given these results, it is not possible to prove the feasibility of this approach and assess its effectiveness in more realistic situations.
Maybe the authors should also discuss how it compares to a recent work arXiv:2309.02292 where the same method is applied.

Requested changes

The numerical experiments need to be done more carefully: 1. describe correctly the experimental setting 2. to consider larger system size and systems with more than one interacting parameters

---

## Round 1 · Referee Report · Anonymous (Referee 2) · 2024-1-19

Strengths

Bridges different fields

Weaknesses

Unclear motivation of the paper

Report

This paper establishes some connection between topological phase characterization and information theory and provides an ML method to estimate quantitatively the entanglement.

The first observation consists in to reinterpret the topological entanglement entropy as an information theoretical quantity which corresponds to the fact that long range correlations related to entanglement can be detected by looking at a topologically non trivial multipartite subsystems. This quantity is basically the Kullback-Liebler divergence between the sub-system's distribution and the Bethe-approximated one, which is normally exact on topologically trivial multipartite system (ie forming a tree of dependencies). If probably implicit in many previous works, this seems to me (being a non specialist of topological entanglement) a clarifying observation of TEE. This at least let them generalize in a straightforward way the relation between the TEE and this conditional information I_n for n-partite systems.

This is then illustrated on a Z_2 gauge theory, by estimating the TEE on Wilson loops of arbitrary size n. For this, they consider a proxy for the TEE (which is still difficult to measure for large n) which is the n-coupling {\cal I}_n of generalized Ising model, encoded into a restricted Boltzmann machine. One of the main results is the equation (51) which allows one to extract the Ising couplings from the parameters of RBM which can be of use in other contexts. Actually I found part IV.C, which purpose is to justify (51), quite lengthy and obscure because it seems to me that the quantum formalism is superfluous since nothing is proposed how to generalize this beyond Z_2. (Actually straightforward way to obtain (51) with binary classical spins is to use simple binary identities like \sum_{s_i}\prod_i (1+s_i s'_i)/2 = \prod\delta(s_i-s_i') and extract the spin s_i from K(\sum_i w_{ij}s_i) ).

At this point I am a bit lost in the motivations of the paper, because the justification for using {\cal I}_n instead of I_n is quite weak, since to my knowlegde no quantitative relation exists in general between the two.
Maybe we might expect some statistical correlations in this specific setting where the targeted coefficient cluster coefficient I_n corresponds to the size of the visible layer of the RBM. It could be interesting to check this more systematically for various Wilson loops sizes n up to typically ~20. In the end a numerical illustration is actually provided, where the entanglement is estimated numerically through {\cal I}_4 on small square Wilson loop with a RBM with 4 hidden nodes which is not very instructive, despite the fact the RBM learns properly the distribution of the subsystem which is to be expected
Considering characterization/identification of topological phases,
(if this is the motivation) I am not sure of what this is of practical use. This point should be clarified. Additionally once the goal of the paper are better clarified, the use of the RBM should be also better motivated.
Since at present its use is illustrated only on Ising subsystems corresponding to Wilson loops, Maybe I am wrong, but I am not sure that the RBM constitutes
the best choice in that case, because in fact on a loop the
inverse pairwise Ising problem can be solved exactly, (see Chertkov Chernyak 2006 for loop corrections, and more specifically Decelle Furtlehner JstatPhys 2016 Sec 5 for the inverse Ising problem on loopy graphs), so that the TEE can be presumably estimated appropriately by the KL distance between the Wilson loop distribution and its Bethe approximate model, which boils down to the loglikelihood of the pairwise Ising model (including the normalization constant so that it is exactly zero in absence of TEE).

Small remarks: there are typos here and there ... as beginning of p 6: "in order expose.."
p9 full sentence: "This allows us to interrogate the RBM ..." should be rephrased to be understandable.
bottom of p10:The the

---

## Editorial Decision

awaiting_resubmission